# Habits, Health and Environment in the Purchase of Bakery Products: Consumption Preferences and Sustainable Inclinations before and during COVID-19

**DOI:** 10.3390/foods12081661

**Published:** 2023-04-16

**Authors:** Agata Nicolosi, Valentina Rosa Laganà, Donatella Di Gregorio

**Affiliations:** Department of Agriculture, Mediterranean University of Reggio Calabria, Feo de Vito, 89122 Reggio Calabria, Italy; anicolosi@unirc.it (A.N.); donatella.digregorio@unirc.it (D.D.G.)

**Keywords:** bakery products, sustainable foods, consumers, exploratory factor analysis, PLS-SEM

## Abstract

The aim of the research is to investigate whether purchasing decisions about bakery products (bread, snacks and biscuits) are influenced by concerns about health, climate change, biodiversity loss and food waste. The exploratory survey was carried out in two successive moments before and during the health emergency from COVID-19. Before the health emergency, face-to-face interviews were carried out using a structured questionnaire. Data were analyzed by factor analysis, reliability tests and descriptive analysis. Structural equation modeling (SEM) was employed to test the research hypotheses. The results of the modeling analysis of the structural equations highlighted that health and the environment represent an important background in the consumer experience of the respondents and influence the attitude and intention to purchase safe and environmentally friendly bakery products. Furthermore, the results suggest that informed, modern and aware consumers have direct and indirect effects on the intentions to adopt sustainable attitudes. On the contrary, the perception relating to the shops where consumers buy bakery products does not always show a significant influence on the propensity for sustainability. During the health emergency, the interviews were conducted online. Families confined to their homes, buying less in stores, have prepared many baked goods manually at home. The descriptive analysis of this group of consumers shows a growing attention to points of sale and the tendency to use online shopping. Furthermore, the changes in the type of purchases and the importance attributed to the need to reduce food waste emerge.

## 1. Introduction

The processes of social transformation underway in recent decades have radically changed the world of food and the agri-food sector [1,2,3]. The COVID-19 crisis has changed people’s daily routines: isolation, remote working, increased sedentary lifestyle and the frequency of meals and snacks, mood changes, anxiety and health concerns have conditioned the choices and reasons for food preferences, the choice of outlets and purchasing methods [4,5,6,7,8]. In the current post-COVID-19 period, the European consumer is looking for quality and healthy foods with natural ingredients to protect themselves from disease, protect the environment and provide sustainability to local economies [9,10,11].

The scientific literature is full of theories concerning the consumer, not only and not so much with reference to aspects of a strictly economic nature (consumption/production) but also to social, psychological and environmental aspects linked to respecting and protecting the environment. Ethics, responsibility, ecological orientation and sustainability are some of the new models that individuals are looking for, also subjecting their food choices to continuous revisions and reflections [12]. Sharing platforms, social media, influencers, attention to food waste, social networking networks, etc., represent a new economic and cultural approach to dealing with food [13,14,15,16,17].

Bread and baked goods are products linked to every moment of the day (meals, sweet and/or savory snacks, parties, sports, etc.) to the daily life of Italians and many populations, and also enjoy a continuous demand for innovative products that reflect the interest of producers in the request for long leavening times and the careful selection of raw materials (unrefined flours, lots of fiber, little gluten). Consumers have renewed interest in preventing food waste and increasing interest in bread in many areas of Southern Italy; people do not throw it away but instead use it for other preparations and/or for other uses. Furthermore, aroma, fragrance and flavor are associated with pleasure and health. Consumers are becoming increasingly attentive to nutritional profiles; even bread and bakery products have become the object of research in terms of well-being, not only physical, but also psychological [18,19].

Even in Italy, food consumption has long since reached a defined phase of satiety; on this basis, the main needs are now satisfied, and the nutritional availability is higher than the physiological needs of the population. The demand for bread and bakery products has also changed: the consumption of bread has decreased, and the requests for a wide range of bakery products have increased. This situation has led to the development of new food consumption patterns and the emergence of new products with specific production and quality standards adapted to the rules that protect the consumer. Today, consumers are more attentive and inclined to make purchasing decisions with greater awareness. They are more informed and prefer food that meets the requirements of food safety and environmental, social and economic sustainability without neglecting the ethical aspects related to the protection of territory, biodiversity, working conditions and, in general, the resources to be handed over to future generations [20]. The new characteristics of the demand (based on a vast range of food products that focus on quality) have been implemented by the players in the supply chain who have adapted to the new market conditions to create products that meet the new needs of consumers [21,22,23].

In light of these new needs requested by the consumer, in addition to guaranteeing the safety, digestibility and palatability of the product, companies aim to use food processing processes that lean toward circular economy production models and the reuse/recycling of by-products and waste as resources to obtain new products. In recent years, food has become increasingly industrially processed [21,24,25]. Companies are also looking to improve shelf life and make it easier for consumers to prepare meals when needed [22]. Product and process innovations that introduce new nutrients and improve the shelf-life of sustainable food and packaging are also aimed at reassuring consumers in terms of ethics and attention to the environment, climate and biodiversity [23,26].

The purpose of this analysis is to investigate the decision-making process of consumers’ choices when purchasing bakery products and, in particular, in relation to the following: 1. The importance attributed to environmental, social and economic sustainability; 2. The attention they pay to health and food safety requirements; 3. Curiosity and the search for information; 4. Preferred points of sale for the purchase of different types of bakery products (bread, snacks and biscuits). The study explores the degree of awareness of a sample of consumers in Southern Italy regarding the attention to food quality and safety and the adoption of virtuous and sustainable behaviors into their lifestyle.

Many researchers have explored a number of factors that influence people’s perception of their inclination toward sustainable food and environmental and social sustainability [20,27,28,29,30]. Some studies have attempted to combine various psychological determinants into a predictive model of behavioral intentions [31,32,33,34].

In the first period of the survey, to analyze the possible choice preferences of the people interviewed face to face, the research questions revolved around the reasons for the choice, such as quality and safety, taste, experience, habits, points of sale and, in particular, the focus was to grasp the critical awareness of the interviewees in terms of environmental and social issues and the importance of the drivers of preference in their choice of store. 

Our hypothesis is that consumer behavior responds differently on the basis of predisposition and inclination toward sustainability.

To confirm our intuition, we applied a methodology that included factor analysis and partial structural least squares equation modeling (PLS-SEM), which has recently received increasing attention in research and practice in various disciplines, such as management, marketing, political and environmental sciences, and other fields.

A separate discussion relates to the online interviews carried out during the lockdown. People were confined to their homes and, especially women, responded quickly and willingly to the structured questionnaire in a simpler and more direct way. The main objective, in this case, was to investigate the changes in terms of family organization during the lockdown in relation to the purchase and preparation of food, particularly bakery products.

The paper is structured as follows: The introductory section is followed by a description of the bakery products sector. Subsequently, the conceptual framework and the research method are outlined. The latter presents the results examined before and after the COVID-19 pandemic, discussion and conclusion.

## 2. Bakery Products

The bakery products sector includes a large and varied set of products, and in 2021 a total production of 1,310,412 tons was recorded in Italy, up by 2.6% from 2020, for a total value of 6310.8 million euros, an increase of 6.7 percentage points compared to the previous year.

The pandemic, war in Ukraine, inflation and energy price increases have led to a marked increase in the price of bread which, according to Eurostat monitoring, grew in August 2022 by 18% on average compared to the same month in 2021. Eurostat notes that the consequences of the Russian invasion of Ukraine (two giants in cereal exports), are at the basis of the widespread increase in prices. The largest increases in the cost of bread in the period August 2021–August 2022 were recorded in Hungary (+66%), Lithuania (+33%), Estonia and Slovakia (+32%). The least-affected countries were instead France (+8%), Holland and Luxembourg (+10%). In this context, Italy ranks below the EU average (+13.5%). To highlight the sensational jump in prices, for a comparative purpose, the European Institute of Statistics recalls that between August 2020 and August 2021, the average increase in the price of bread in the EU was 3%. 

However, despite the difficulties, bread, a food symbol of the Mediterranean culture and diet, has not lost its centrality in the daily life of Europeans and confirms itself as a refuge food, even in times of crisis.

In fact, bread unites the nations of the Mediterranean “making/(producing) bread” helps to strengthen the image of a united and plural Mediterranean that, still today, perpetuates traditions and cultural identities.

The model of bread consumption is diversified; from the 2020 data, the highest levels are recorded in Romania, with 88 kilos per capita per year, followed by Germany (80 kilos every year), followed by the Netherlands (57 kilos), Poland (52 kilos), Spain (47 kilos), France (44 kilos) and the United Kingdom, with 43 kilos per capita, and lastly, Italy, which consumes 41 kilos of bread each year.

Among the products protected by the trademarks of the European Union we find in category 2.3: “Bread, pastry, cakes, confectionery, biscuits and other baker’s wares”, relating to bakery products, there are 101 certifications, and the sector is constantly evolving. In particular, the EU site “eAmbrosia: is a legal register of the names of agricultural products and foodstuffs, wine, and spirit drinks that are registered and protected across the EU. It provides direct access to information on all registered geographical indications, including the legal instruments of protection and product specifications”. The Italian segment has four Protected Designation of Origin (PDO) products and 12 Protected Geographical Indication (PGI) products (Table 1).

Bakery products obtained by cooking a leavened dough prepared with floured wheat, water and yeast, with or without the addition of common salt, is called “bread”. The bread can be sold loose by weight or pre-packaged and pre-wrapped.

The generic term “bakery products” refers to foods obtained by cooking leavened dough, in which the basic ingredients are flour, water, yeast and salt. In Italy, these products fall within a specific economic category defined in the ATECO, and the products that fall within it can be traced back to six sub-categories: biscuits, crackers, baked sweets, focaccia, pizzas and cakes [35]. For bakery products, there are two classes involved, class 10.71 and 10.72. According to the international ISIC system, it is Class 1071, Manufacture of bakery products; in NACE Rev2, these activities are included in class 10.71 [36,37].

Most of these products are part of the traditions and history of individual countries, while others are the result of contamination and globalization that is found both in uses and in food. Increasingly, the current trend is to replace wheat with various grains, corn, fiber and other foods, which include various proteins. This has caused a substantial increase in the production and consumption of alternatives such as crackers, rusks, bread sticks and sliced bread; especially in countries with advanced economic development, there is a tendency toward the contraction of bread consumption [38].

These are longer-lasting products that are much more compliant with consumption which, in a more frenetic lifestyle, often does not facilitate the use of foods that must be purchased on a daily basis. After all, specific surveys on consumer behavior show that fresh bread is among the main foods subject to food waste, and this is due to the need for daily consumption [39]. Bread and bakery products form an essential component of the human diet worldwide [19] and constitute a very important segment of the global food industry [40]. The interest of scholars is great, with very diversified research on bakery products and alternatives to them [41,42], including the diffusion of functional bakery products [43,44,45,46,47].

## 3. Survey Method

### 3.1. Work Plan, Data Collection

The data were collected before and during the COVID-19 pandemic. In 2017, before the health emergency, the sampling scheme envisaged a random system for the identification of the consumer. The data collection was carried out face-to-face with consumers who were willing to be interviewed and intercepted in Southern Italy (especially in Calabria and Sicily). A semi-structured questionnaire with free and/or preformulated answers was used. In order to meet a sample with non-homogeneous characteristics and intercept consumers with different purchasing methods and abilities, the questionnaire was administered in particularly crowded places, such as bus terminals, railway stations and ports, main roads, retail outlets, mass-market retailing (MMR), local markets and local food and wine events. The forms prepared were tested on various occasions, allowing for modifications to be made to the questionnaire to better adapt and perfect it for the purposes of the research. In total, 742 questionnaires were completed, of which 22 were discarded, with a valid response rate of 97% (720 valid responses).

The questions in the questionnaire aimed to identify the following:Socio-demographic characteristics of the interviewees;Information on the consumption and type of bakery product purchased (artisanal/local or industrial)Information on the place of purchase of the products;Frequency and quantity purchased;Description of buying behavior.

In this last section of the questionnaire, particular attention was paid to consumer awareness of the sustainability implications of the products purchased. The questions concerned the conditions of choice on food quality and safety, taste, experience, habits and attention to environmental and social issues that can influence preference. In particular, consumers were asked if they pay attention to the origin of the raw materials, the information on the label, the importance of biodiversity, packaging, certification marks, nutritional information, the reputation of the manufacturing company, traceability and tracking, the clarity of the transformation procedures, the possibility of purchasing online and the adequacy and correctness of advertising.

This study used closed-ended questions that were prepared by taking into account the research of studies present in the literature and that were useful for measuring the tendency and sustainable inclination of consumers on the purchasing choice processes. The questions were arranged to collect both binary (yes/no) answers and multiple choice. The scoring is based on a five-point Likert scale, with responses indicating the following: 1 = not relevant, and 5 = very relevant. The compilation of the questionnaire was carried out by the authors and by collaborators/researchers educated and trained in field research [31,48,49].

The extent and complexity of the survey tool used have made it possible to obtain an abundant mass of data and information. The database was created through the use of SPSS V20 and PLS3 software. In total, 35 variables (previously coded) were entered, following the order of the questions posed in the questionnaire, in order to create a database aimed at detecting consumer behavior.

In 2020, the interviews were carried out online, and 474 consumers scattered across various regions of Southern Italy were intercepted. This approach has allowed us to examine some attitudes regarding the consumption preferences of bakery products.

We structured the questionnaire administered online in 2020 during the confinement period caused by the COVID-19 pandemic in a simpler way with a specific set of predefined questions in order to obtain greater participation. The data were collected from March to April 2020. In that period, the state of the pandemic was in force in Italy.

In Italy, we were going through the so-called phase 1, which lasted from February to May. A distinctive element of this phase was home isolation: Italians were obligated to stay at home, only being able to go out for the exceptions of proven work needs, situations of primary necessity or health reasons. During these months, the use of electronic devices and the internet became increasingly frequent, and the power of sharing through social networks increased (the hashtag #stayhome has been typed in by Italian users 245,000 times). Combining the restrictions of the lockdown and the possibility of resorting to a remote survey methodology, we have formulated an online questionnaire that can be filled out online and which is written in an intuitive language that is accessible to everyone, launched using the Microsoft Forms platform. Of all the social networks used for sharing, including the WhatsApp messaging application and the Instagram platform, the one that played a crucial role was the Facebook portal.

The questionnaire was, in fact, published on the electronic bulletin boards of various groups, obtaining a particularly significant response from women, representing approximately 75% of the entire sample of users. This clear female majority is dictated by the fact that these groups have been used a lot by women to exchange useful advice in various fields: domestic/culinary, above all.

From a methodological point of view, the online survey is configured as snowball sampling. In addition to being a forced choice by the pandemic, the choice of this type of sampling also made it possible to reduce the time and costs of the investigation. The technique allows for the acquisition of data through one’s acquaintances together with the use of social networks and in which each study subject recruits other subjects among his/her acquaintances [50,51], a sort of chain reaction.

The questionnaire was relaunched and advertised several times through online social networks. A total of 486 questionnaires were completed, of which 12 were rejected, resulting in a valid response rate of 97% (474 valid responses).

### 3.2. Methodological Approach

The methodological approach adopted for this research followed two directions before and during the COVID-19 pandemic, as illustrated in the data collection phase.

The data collected before the COVID-19 pandemic were processed using multivariate analysis techniques. To identify which drivers move consumers in the process of choosing the baked goods to be purchased/consumed, an exploratory factorial analysis (EFA) was carried out based on the analysis of the main components (PCA). The model aims to reduce the number of predictors in the factor dimensions by minimizing the loss of variance [50,52]. The analyses highlight latent factors. The reliability of the model was evaluated using two different tests: Kaiser–Meyer–Olkin (KMO) and Bartlett’s spherical test [53,54]. Subsequently, the analysis of the models of structural equations was carried out (PLS-SEM) to test the hypotheses concerning the relationship between the determining factors and the sustainable behavior of consumers [32,33,55,56,57].

In the context of the literature concerning food consumption, many studies have applied the Theory of Planned Behavior (TPB) [58] to investigate the intentions and behaviors of consumers. The predictive power of TPB has been successfully applied in many research fields on food consumption, such as those concerning healthy behaviors [59,60], healthy eating, pro-environmental behaviors and the consumption of organic foods [55,61,62].

The modeling of structural equations (SEM) has become the methodology of choice for many researchers who study complex relationships between latent constructs, such as in marketing, consumer choices, and other fields. Its ability to evaluate complex measurement models and structural paths that involve many variables and construction levels has allowed researchers to investigate complex relationships that previously could not be easily examined. As highlighted by Lei and Wu [56], SEM represents an advanced version of the general modeling procedures and is used to evaluate “*if an hypothesized model is consistent with the data collected to reflect [the] theory”* [56].

The theoretical framework in this study resumes the model widely used by Partial Least Squares Structural Equation Modeling (SEM). The model allows for the explanation of more statistical relationships simultaneously [57] to understand the relationship between latent constructs (factors), generally indicated by different sizes, and adopts a confirmation approach after examining the data with exploratory analysis, and latent factors are explained through dependence reports [63]. SEM provides a single complex model that includes various dependence and independence relationships between constructs. Recently PLS-SEM) has become rather popular among researchers/scholars [64,65,66,67,68].

In this study, PLS is used as the data analysis tool for the research model. PLS is an SEM analysis technique based on regression analysis, which is a statistical method derived from path analysis. Using PLS analysis, it is possible to examine both the measurement model of the research instrument and the structural model of the research component [65,69,70,71]. Furthermore, PLS can also be used with a smaller sample size. Compared to other analysis models, PLS requires multivariate constant assumptions and is better at predicting and has greater flexibility [67,68]. The PLS analysis software used in this study is SmartPLS3 and uses the blindfolding procedure to check the significance of the paths in the structural model.

### 3.3. Research Hypothesis

The conceptual framework for this study is presented in Figure 1, which illustrates the potential reasons that influence the behavioral intention to identify a sustainable inclination of consumers. The perceptions of risks concerning climate change, the loss of biodiversity and food safety and the perception of the benefits related to environmental and social respect and health are considered predominant factors that contribute to the adoption of sustainable behaviors. When individuals or groups of consumers perceive that there are potential benefits from attentive behavior to the environment and society, they are more inclined to adopt sustainable food choices, especially for widely consumed products, such as baked goods.

The favorite points of sale in terms of consumers, availability and the possibility of online purchase also come into play in these choices. When consumers perceive that clean and sustainable production and transformation methods could lead to major well-being in society, that technology can improve the quality of life, that it is worth trying new food products and that food is important for a healthy lifestyle, then consumers have a greater possibility of perceiving greater benefits and fewer risks to their health and the environment, accordingly, they adopt adequate choices and behaviors.

#### 3.3.1. Consumption Experience (CE): Health Awareness and Environmental Awareness

Health awareness and well-being expectations are thought to be key variables influencing buyer intent [59,62].

Food safety, nutritional aspects, the flour and raw materials used, production and marketing are important elements for consumers. Therefore, health consciousness evaluates an individual’s readiness to take health-related actions and represents the most significant reason for the conscious purchase and the consumption of food with safety requirements related to bread and baked goods and is an important predictor of the intention to purchase such foods.

Furthermore, hedonistic variables and sensory emotions, such as aroma, flavor, freshness, crunchiness and product appeal, are important predictors of purchase intentions and highlight a growing interest in sustainable food and health-oriented lifestyles [72].

Environmental awareness also plays an important role with regard to the sustainable inclination and intention to buy environmentally friendly food. It refers to the *“emotional point of view of individuals on the environment”* [31]. Issues relating to environmental friendliness are applied by consumers to food in relation to the origin of raw materials and information concerning production, processing, marketing, etc. The environmental awareness of consumers encourages their positive attitude toward the purchase of foods that meet environmental sustainability requirements.

Awareness of health and environmental aspects represents the consumer’s wealth of knowledge and the consumer’s experience. These are subjective norms derived from consumption habits that are influenced by social stress from others, such as friends and family, causing individual motivation to engage in and respect group behavior [31]. The subjective norm is a critical factor affecting social influences and behavioral intentions. Previous analyses have also revealed a significant relationship between attitude and subjective norms [73]. If people who are meaningful to consumers offer positive opinions and attitudes toward sustainable food patterns, consumers will be more likely to have a positive intention to buy food with sustainability requirements.

In this regard, a study of two developing nations, namely Tanzania and Kenya, found that health consciousness, along with personal attitude and individual norms, are important parameters influencing the intentions behind consumer purchases [31].

Based on these arguments, the following hypothesis is formulated.

**Hypothesis 1 (H1).** 
*The consumer experience (CE) and subjective norms has a positive and significant impact on consumers’ inclination towards a model of safe, sustainable, ethical and responsible consumption.*


#### 3.3.2. Informed Consumers 4.0 (IC) and Choice of Points of Sale (St)

In the past, consumers were mainly informed about food by newspapers, magazines and television advertisements, and the main motivation for purchasing organic food is that individuals believe it is more beneficial for their health. However, when consumers buy products in the market, they cannot obtain complete information [69]. To overcome this information asymmetry, social media has become an indispensable part of the promotion of food (organic, with certifications, ethical, etc.), also providing information within the category of bread and bakery products.

In fact, in the last 10–15 years, consumer society and the globalized world have strongly influenced the changes in lifestyles and consumer habits. The Internet and the use of social media offer information that has influenced important changes in people’s behavior. Currently, social networks (including blog forums) represent a virtual space in which users can create and share multimedia content and interact with other users interested in the same topics. Consumers can now participate in discussion groups to ask for advice, information and assistance in the decision-making process before purchasing [13,74]. The development of relationships between consumers within social networks results in the formation of social and even emotional ties.

Consumer 4.0 is a modern, informed and aware consumer. They seek innovation and pay great attention to health. They tend to buy certified products but with a critical sense. They represent the fusion between tradition and contemporaneity, respect the history of their territory and also project into the future.

Based on the observations highlighted, we hypothesized that informed consumers 4.0 (CI) have a direct effect on sustainable inclinations.

**Hypothesis 2 (H2).** 
*Informed consumer 4.0, it deals with a mindful, innovative and digital consumers who have a positive and significant impact on inclination towards a model of safe sustainable, ethical and responsible consumption.*


As for the choice of points of sale, that is, where to buy bread and bakery products, consumers have very diversified habits and, in many cases, they buy both in bakeries and in supermarkets, artisanal bread and/or packaged sandwich bread and thus also biscuits and snacks.

Konuk [75] states that the role of the store image, in relation to perceived quality, trust in the store and in the seller and perceived value, influences consumers’ purchasing intentions. In fact, trust is defined as *“the consumer’s expectation for the reliability of the services provided and on the fulfillment of the promises by the supplier”* [76]. The retail market has been facing stiff competition, and companies are struggling to differentiate themselves in the market. In this context, the image of the store is one of the most important distinguishing features that provides a substantial advantage for retailers. It represents the set of perceptions of a consumer with respect to a store with reference to different attributes. In other words, the point of sale (shop or open-air market) is experienced as a space for socializing and creating an atmosphere of trust in the quality of the goods and the availability of the shop’s staff [8,75]. Loyalty is like *“a willingness to rely on a trusted trading partner”* [75]. The trust theory of commitment emphasizes that trust is a prerequisite for maintaining long-term relationships with the firm. Based on the observations highlighted, we hypothesized that the choice of store made by consumers has a direct effect on the inclination toward sustainability.

**Hypothesis 3 (H3).** 
*The choice of stores has a positive and significant impact on the inclination towards a safe, sustainable, ethical and responsible consumption model.*


Furthermore, two research hypotheses concern informed consumers 4.0 (CI). The first concerns the experience and the Experience of Consumer (CE) and pertains to the influences that come from the information obtained from social media, etc.

**Hypothesis 4 (H4).** 
*Informed consumers 4.0 have a positive and significant impact on the perception of choice of stores where you can buy baked goods.*


The second hypothesizes the influence of informed consumers on the behavior and choice of places to purchase.

**Hypothesis 5 (H5).** 
*Consumer informed 4.0 have a positive and significant impact on the perception of consumer experience.*


In this context, information concerning how the store’s image, perceived quality, seller trust and private label also influence consumers’ purchasing intentions for food and bakery products. The hypothesis formulated is the following:

**Hypothesis 6 (H6).** 
*The choice of stores has a positive and significant impact on the consumer experience.*


## 4. Results

### 4.1. The Sample Data before and during COVID-19

The socio-demographic characteristics of the subjects interviewed in the two periods of investigation are illustrated in Table 2, which distinguishes between those who were interviewed in face-to-face mode before the pandemic and those who were interviewed during the COVID-19 period when the lockdown was in force in Italy (home insulation).

In the pre-COVID-19 period, the sample interviewed face to face is characterized by consumers of a medium–high cultural level (34% are graduates), with a slight majority of men (50.3%). The average age is 43.86 years (from a minimum of 28 years to a maximum of 89). The standard deviation is 16.83%.

By entering greater detail of the socio-demographic characteristics of the sample interviewed, it can be observed that the most represented age class is between 30 and 49 years (34.3% of the sample), those who are more than 49 years old are 38.5%, and young people aged 18 to 29 are 27.2%. Schooling highlights a greater presence in graduate subjects (47.4%). Graduates represent 34%. Most interviewees say they receive a medium–low (43.3%) or medium–high (38.3%) family income and 12.4% have a low family income. In a few cases, their income is high (6%), and the most represented number of family members is families made up of four people (30.7%), while the largest families (equal to or greater than five people) represent 4.5% of the sample. The interviews carried out during the COVID-19 period show a very different situation. First of all, the greatest presence of women due to their greater presence and participation on Facebook and social groups in general emerges. As was to be expected, there is a lower average age (39 years). About 68% of the participants are in an age range from 18 to 49 years old with average higher schooling; 50% graduated, and 42% are post-graduates (of which 6.8% attended post-graduate courses). Family income is mainly medium–low (51%), followed by medium–high income (35.6%) and low (11%).

The shopping habits we observed demonstrate a clear distinction in the purchase of bread, snacks and biscuits both before and during the COVID-19 period.

In Southern Italy, the purchase of artisanal bread prevails (about 70%), while packaged bread is preferred in 14% of cases. Furthermore, some interviewees buy one or the other type of bread indifferently (16%). People prefer to buy it mainly at the bakery (63%), the hypermarket (23%), in the city market or from small retailers (14%).

The opposite situation is found for snacks and biscuits, whose prevailing preferences are for packaged products (38% and 44%, respectively), purchased mainly from MMR. A group of consumers declares that they prefer artisanal snacks (25%) and artisanal biscuits (14%) purchased at the bakery. Finally, a group of consumers interviewed purchased both artisanal and industrial snacks and biscuits (32% and 37% of cases, respectively). Finally, about 5–6% of those interviewed do not buy them or did not answer the question.

During the health emergency due to COVID-19, there was an increase in the online purchase of packaged baked goods in the face of high home preparation of baked goods. During the lockdown, families prepared bread, focaccia and snacks and sweets at home in 75%–85% of cases. Only 5.3% stated that they had not prepared baked goods at home. In the remaining cases, they were regularly purchased in stores.

### 4.2. The Results of the Multicriterial Analysis Applied to the Data Collected before COVID-19

#### 4.2.1. Factorial Analysis

Factorial analysis was applied to 16 variables. The value of the KMO test is 0.802. To be considered reliable, the value of the KMO test should be between 0.5 and 0.7. At the end of this first analysis, four components were extracted that identify four groups of latent factors that explain 57% of the total variance (see Appendix A).

Table 3 reports the loads of the rotated components that allow for a description of the predictive models synthesized of preferences by reducing the multidimensionality of the variables. In this way, it was possible to associate each component with the main drivers of the choice of preferences.

As regards the first component (17.5% of the variance explained), it can be identified as “sustainable attitudes/inclination” (factor 1); the variables involved are the expiry date (0.78) and the information on the label (0.74), the reputation and ethics of manufacturing companies (0.68), sustainable and differentiated packaging (0.62), the raw materials used and respect for biodiversity (0.59), and finally, local km0 products (0.195).

The second component, which represents 15.1% of the total variance explained, can be described as “consumer experience” (latent factor 2) since many attributes included in this component are directly or indirectly related to quality and safety (0.63), at a fair price, both for consumers (0.62) and for producers (0.59); also, the interviewees hold a lot of importance on the clarity and correctness of the transformation techniques (0.52) and the importance attributed to PDO, PGI brands (0.48).

The third component, whose variance explained 14%, can be defined as “stores”, and concerns the importance of the store.

Finally, the fourth component, which represents 10.3% of the variance explained, describes the most current thrusts, such as online purchases (0.77) and the demand for transparent and correct communication, advertising and marketing (0.62) and identifies a group of consumers, “Consumers informed 4.0”.

#### 4.2.2. The PLS-SEM Model

The PLS–SEM path analysis algorithm estimates the standardized partial regression coefficients in the structural model after approximating the measurement model parameters. After carrying out the CFA (confirmation factorial analysis), the validation of the study model and hypotheses took place through the modeling of the structural equations (SEM) following the indices of the model adaptations [67]. The outcome of CFA shown in Table 4 indicates that the standardized loads of all the remaining elements in the measurement model were above the acceptable cut-off level. The standardized charges of the elements that make up the measurement model are between 0.622 and 0.950. The quality of the measurement model was tested by examining the indicator reliability, internal consistency, convergent validity and discriminant validity. Indicator reliability was assessed by exploring the standardized loadings of items with their respective construct. According to the indications of Dash and Paolo [57,77], items should be retained in the measurement model only if their standardized loadings are equal to or greater than 0.6. Since the model of the Loadings of the CE5 and CE6 items were lower than the recommended threshold value, they were removed from the measurement model and from further analysis, which led to values that were in the acceptable interval for all three indices of the internal coherence of the construct. The validation of the study model and hypotheses took place through the modeling of the structural equations (SEM) following the indices of the model adaptation recommendations [67]. The values of the correlation matrix of the constructs analyzed are examined in Figure 2 and in Table 4.

The study used PLS-SEM to study the relationships between the constructs using SmartPLS3, an effective statistical tool that deals with complex models and is suitable for small and large datasets [78].

The first fundamental step in the development of the measurement model was to evaluate the convergent validity through the following criteria: the loading factors, composite reliability and the average variance extracted (AVE). Subsequently, all of the other indicators were taken into consideration, as in the literature by many authors.

As shown in Table 4, the standardized loading of the items was greater than 0.6 for all factors, as recommended by Chin [79]. In particular, the elements with a loading value equal to or greater than 0.7 were significant. The composite reliability values (CR) of all factors exceeded the recommended value of 0.7 [71,80,81].

An AVE value of 0.50 or greater is considered acceptable because it indicates that the variance shared between a construct and its elements exceeds the variance of the measurement error [78]. However, in our study, the values of AVE have a score greater than 0.5, all except one, which is slightly lower (consumption experience, with an AVE equal to 0.491). However, these data did not affect the results and validity of the applied model.

In fact, the Fornell–Larcker criterion and the Heterotrait–Monotrait (HTMT) correlation ratio are valid (Table 5 and Table 6).

As for the Fornell–Larcker criterion [33], it states that the square root of the AVE of each construct should be greater than its highest correlation with any other construct in the model. The results presented in Table 5 indicate that each construct shares more variance with the elements allocated to it (bold values on the table diagonal) than with the remaining constructs in the model, thus confirming that the requirements of the Fornell–Larcker criterion are satisfied.

In addition, the HTMT correlation report is an alternative approach to evaluate the discriminating validity in PLS-SEM. It has been reported that this method has higher performance than the Fornell–Larcker criterion. The HTMT should be lower than 0.85 (a stricter threshold) or 0.90 (a more lenient threshold), or significantly smaller than 1 [82,83]. As shown in Table 6, all of the HTMT values are lower than 0.85, thus indicating a good discriminating validity.

The results of the SEM path are shown in Table 7, which shows that the T-Values for the routes of H1 (15,292), H5 (12,796), H2 (5096) and H6 (2540) are higher than the value standard. The hypothesis that the consumer experience, informed consumers 4.0 and the choices of the stores have a positive influence on the sustainable inclination of consumers when they buy baked products. However, the perception of the bakery product stores does not show a significant influence on sustainable inclinations (both directly and through informed consumers); consequently, hypotheses H3 (0.849) and H4 (1620) are rejected.

The blindfolding procedure was applied to verify the significance of the paths in the structural model of the effect sizes (F^2^), the Q2 construct and the multicollinearity model. As indicated by the reference indices [71,82], the values of Q2 are above 0. As far as F^2^ is concerned, it highlights and confirms what has already been stated by the supported or rejected paths (Table 8 and Table 9).

### 4.3. Consumption and Choices of Agri-Food Products during the Lockdown: Changes and Trends for Bakery Products

The COVID-19 emergency has profoundly changed various aspects of the life of Italians. Consumption dynamics and purchasing behavior have led to changes that did not seem possible before. The new eating habits that emerged from the survey have found a place in Italian lives: preferring quality over quantity, rewarding small producers and short supply chains by giving a different value to food waste and sustainability. The interest in the new Italian routine and the variation in consumption dynamics were examined to capture the changes in aspects relating to consumer habits. We have detected changes in both pre-COVID-19 and during COVID-19, with reference to the points of sale where shopping is done, the frequency of purchase, the type of baked goods purchased and the possible replacement of the latter with homemade products. The attitude toward food waste, the possible tendency to reuse food leftovers and attention toward environmental sustainability were also noted. In this context, bakery products have had particular importance because they are among those that have been made at home in families to a greater extent.

Based on the ISMEA data, as a result of the emergency, consumers’ attitudes are changing both with respect to the products purchased and the sales channels used. There is a trend toward the procurement of preservable products (pasta, rice, fish preserves, tomato preserves, etc.) to create household stocks and to make bread, pizza and desserts at home.

The cereal supply chain was the one that presented the most critical issues during the lockdown: at first, the run-up to the supply of flour and pasta caused a sudden peak in demand. Subsequently, slowdowns in imports caused difficulties in ensuring an adequate rate of national self-sufficiency. This difficulty in finding these products led to an increase in prices, in particular, wheat, both soft and durum. This situation lasted until the end of April, and then decreased from May onwards: both the prices and the demand for cereals stabilized, and the self-sufficiency difficulties gradually resolved.

With reference to the other sectors and production chains, increases in the consumption of fruit and vegetables were recorded during the months of the lockdown. Greater purchasing preferences were directed toward more easily storable fruit and vegetables, such as apples, kiwis, cabbage, dried legumes and potatoes, as well as frozen vegetables. During the quarantine, with the greater availability of time for preparing meals, there was a strong increase in the purchase of fresh, full-range and home-prepared products. Conversely, there have been decreases in the purchases of fresh-cut products (that is, packaged and ready-to-eat fresh vegetables).

The dairy supply chain was penalized by a production surplus of cheese and dairy products, mainly due to the closure of the Ho.Re.Ca channel (the main outlet channel for many companies, Hotellerie, Restaurant, Catering). However, in the months between February and June, there was an increase in the purchase of UHT milk by consumers and packaged hard cheeses at large-scale distribution, as they are characterized by greater shelf lives, which was to the detriment of fresh milk and soft cheeses. Beef and sheep meat had a fluctuating trend during the period. The poultry supply chain (chickens, turkeys and eggs) reacted better when compared to the other meat supply chain after the spread of COVID-19.

Table 10 and Figure 3 show the shopping choices at the points of sale before and during the pandemic declared by the consumers interviewed. As can be seen, a very clear change emerges: the small shop close to home records an increase in the percentage variation of +222%, followed by the online market, with +40%. The direct channel with farmers registers +10%. The closure of open-air markets during the period of restrictions shows the greatest percentage reduction (−100%), followed by super and hypermarkets (−22%), which, however, maintain just over 50% of the preferences.

In choosing the procurement channel, the most evident change is the tendency to use online shopping. According to the ISMEA data, in the same period, Italy experienced exponential growth in just a few weeks, causing the delivery system to go haywire (up to +160% on an annual basis).

A direct consequence of what has been highlighted is the decrease in the frequency of purchases (also, these data are explained in Table 10 and Figure 4). Before COVID-19, people went shopping several times a week or several times a day.

The way Italians carry out their shopping has been turned upside down: users who shop for food once a week represent as much as 80%, against 25.5%% in the pre-pandemic period (an increase of 55%). The percentage of citizens who shop two or three times a week has decreased, reaching 18%. At the same time, only 2% maintained the habit of shopping four times a week in the post-pandemic period. In addition to the frequency with which to shop, with the lockdown, Italians have changed their preferences toward what they put in their cart.

The obligation of home isolation due to the need to avoid gatherings has favored the trend toward the purchase of products with prolonged durability and usability in many cases as raw materials for homemade food preparations, but also canned and frozen foods have been placed in the cart. In total, 46% of the respondents said they increased their purchases during the lockdown.

Milk, butter and yoghurt are mostly purchased by 67.30% of users. These products have played an important role during the pandemic as they are the basic ingredients for many dishes, both savory and sweet.

Baked goods and breakfast items, such as biscuits and snacks, are purchased by a third of consumers. However, the most interesting aspect is that as many as 76.4% said they prepare sweets and biscuits for breakfast at home. Furthermore, among the baked goods that were mostly prepared at home in families during the lockdown, there are also bread (84.2% of cases), pizzas, focaccias and savory snacks (79.3%). Only 5.3% stated that they had not prepared baked goods at home.

The survey also made it possible to detect the attention paid by the interviewees to the origin and certifications of various products. A group of consumers declared that they continued to purchase organic food products (14.2%), proximity products at Km0 (13.1%) and products with PDO, PGI, Controlled Denomination of origin (DOC)-certified brands (11.4%).

Finally, consumers show increased attention to food waste. In total, 52% of consumers have increased their commitment compared to the pre-COVID-19 period, and 41% said they maintained their attention, only 7% stated that they had reduced their commitment.

Sensitivity to environmental sustainability and the high reuse of food leftovers are declared to be 90%.

## 5. Discussion and Conclusions

It is increasingly evident that consumer preferences for consumption choices are conditioned not only by liking the products but also by the knowledge that the consumers have [84]. An increasing number of these consumers choose not only what they like but, above all, what is good for them [85], consciously or unconsciously changing their consumption decisions. This is visible from the diffusion of products made with unrefined flour, whole-meal bread and biscuits and products with low gluten content. Precisely this last point seems to be linked to the problems connected to intolerance or allergy to gluten; once almost unknown, it is very widespread among the population today. Basically, the food industry is increasingly sensitive to nutritional aspects, even for traditional products, such as bread and bakery products, a trend that is characterizing new consumption patterns and also new products [42,86,87], aspects that were increasingly accentuated by the pandemic crisis and the greater attention given to health and food safety [88]. In fact, consumers are increasingly paying attention to the sustainability of production processes in support of the local economy and companies [89]. They show a growing interest in the purchase of typical products of the territory of origin and aim to reward the ethical behavior of companies, traceability and food safety [90]. As a result, producers have begun to adapt to new needs to meet consumer demands and transition toward food systems that require a radical transformation in light of the Sustainable Development Goals.

The results of this study suggest that the intentions of Southern Italian consumers to adopt sustainable inclinations in the purchase of bakery products derive from a complex decision-making process involving various factors, such as personal consumer experiences and marketing communication and concerns about health, climate change and the environment.

The analysis developed with PLS-SEM on interviews before COVID-19 tested the hypotheses regarding the relationship between determinants and sustainable consumer behavior. The hypotheses of H1 and H2 have been ascertained, and the results show that there is a positive relationship between the inclination for sustainable consumer behavior and the attitude and beliefs they manifest when purchasing food and bakery products. In fact, this survey shows that consumers who declare attention toward green experiences and those IC are positively influenced by a new, more responsible and more coherent consumption model with green trends. This implies that even when they consume bread and baked goods, they have responsible behavior toward the environment and an awareness of health. Additionally, in the hypotheses of H5 and H6, consumers have a positive attitude toward the aspects related to CE, research on the label, the origin of the food, biodiversity, local food, the importance of recognizing a fair price for farmers and the traceability and reputation of the manufacturing companies. However, hypotheses H3 and H4 are rejected because when consumers choose the point of sale where to buy bakery products, they are not always attentive to the sustainability factors highlighted by the attention gained with the consumer experience, and the same goes for informed consumers who are very selective about the point of sale.

With the COVID-19 pandemic, the virtuous behaviors of consumers seem confirmed, and we seem to be able to say that the state of uncertainty experienced has influenced sustainable behavior and the inclination toward this behavior.

In the coming years, it will be possible to verify whether the behavior toward environmental values and healthy and sustainable consumption will persist [91,92].

The implications of this study are linked to the tools it can provide in terms of knowledge and opportunities in the new economic context for producers, retailers and the distributors of bakery products and foods, in general, to develop adequate green marketing strategies with a message that should be explicit and as detail-oriented as possible concerning sustainability, including the production process and how it positively affects the health of consumers.

The main limitation of this research is that the interviewees were geographically located in Southern Italy; therefore, the results may not reflect the entire nation’s intention to purchase sustainable foods, including bread and bakery products. The consumer trend is shifting toward sustainability and health to maintain health as population aging is on the rise; therefore, further research should follow market trends to determine what future consumers will need. Future studies should also be replicated in the same areas and/or in other geographical contexts or countries to better highlight the changes caused by the pandemic.

## Figures and Tables

**Figure 1 foods-12-01661-f001:**
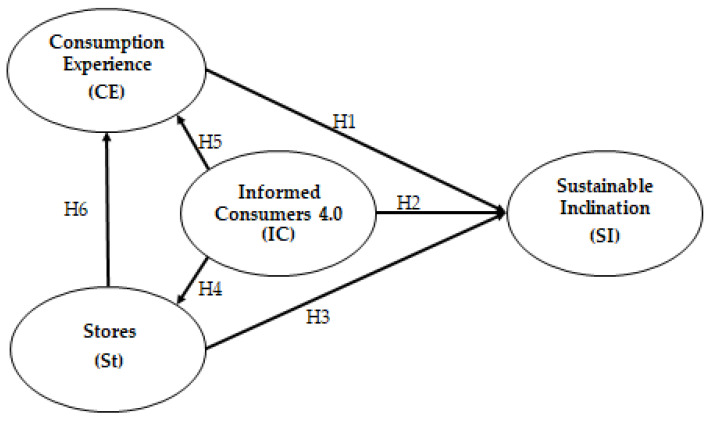
Conceptual model hypothesized in analysis. Source: Authors elaboration from data analysis in Smart-PLS3.

**Figure 2 foods-12-01661-f002:**
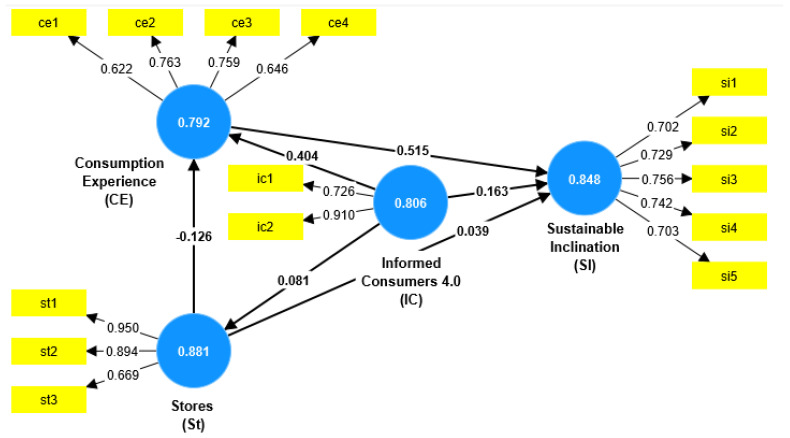
SEM (PLS algorithm). The values in the constructs are Composite Reliability (CR) Source: Authors elaboration from data analysis in Smart-PLS3.

**Figure 3 foods-12-01661-f003:**
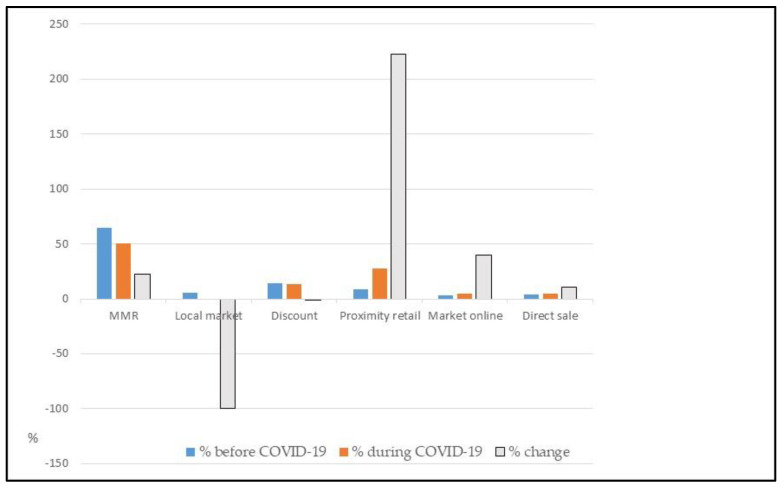
Changes in choosing the stores before and during COVID-19 Source: The authors.

**Figure 4 foods-12-01661-f004:**
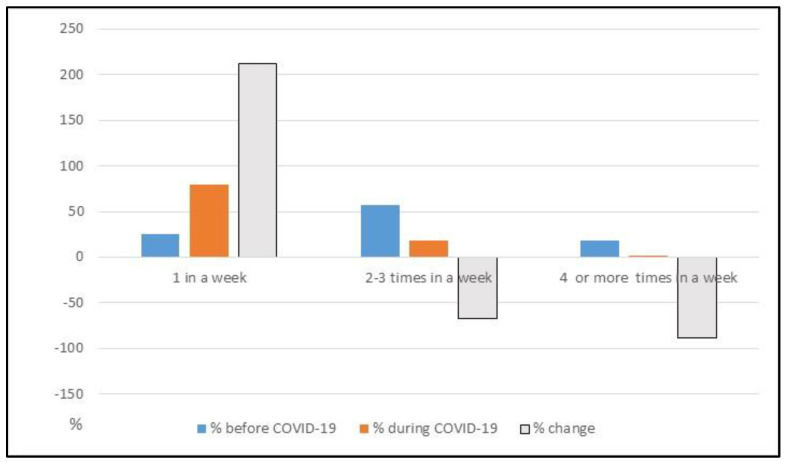
Changes in the weekly spending frequency before and during COVID-19. Source: The authors.

**Table 1 foods-12-01661-t001:** Class 2.3. Bread, pastry, cakes, confectionery, biscuits and other baker’s wares. Product type Food, PDO and PGI in Italy (Data of January 2023).

File Number	Name	Status	Date *
PDO-IT-01016-AM01	Pane Toscano	Published	29 September 2022
PGI-IT-02467	Pampepato di Terni/Panpepato di Terni	Registered	23 October 2020
PGI-IT-02392	Südtiroler Schüttelbrot/Schüttelbrot Alto Adige	Registered	24 July 2020
PDO-IT-01016	Pane Toscano	Registered	4 March 2016
PGI-IT-01290	Cantuccini Toscani/Cantucci Toscani	Registered	26 January 2016
PGI-IT-01323	Pampapato di Ferrara/Pampepato di Ferrara	Registered	8 December 2015
PGI-IT-0944	Focaccia di Recco col formaggio	Registered	14 January 2015
PGI-IT-1067	Piadina Romagnola/Piada Romagnola	Registered	4 November 2014
PGI-IT-1101	Torrone di Bagnara	Registered	14 August 2014
PGI-IT-0795	Panforte di Siena	Registered	22 May 2013
PGI-IT-0666	Ricciarelli di Siena	Registered	19 March 2010
PDO-IT-0577	Pagnotta del Dittaino	Registered	18 June 2009
PGI-IT-0372	Pane di Matera	Registered	22 February 2008
PDO-IT-0136	Pane di Altamura	Registered	19 July 2003
PGI-IT-0120	Coppia Ferrarese	Registered	18 October 2001
PGI-IT-1553	Pane casareccio di Genzano	Registered	25 November 1997

Source: eAmbrosia is a legal register of the names of agricultural products and foodstuffs, wine and spirit drinks that are registered and protected across the EU. * Last registration/modification Date.

**Table 2 foods-12-01661-t002:** Descriptive characteristics of the sample of consumers interviewed.

Variable	Before COVID N° 720 Interviewed Face to Face	During COVID N° 474 Inteviewed Online through Social Media
Gender		
Male	50.3%	74.7%
Female	49.7%	25.3%
Age: average; SD Min; Max	Average 43.86 years; SD = 16.83 Min 18 years; Max 89 years	Average 39 years; SD = 17.4 Min 18 years; Max 76 years
Age class		
18–29 years	27.2%	32.7%
30–49 years	34.3%	35.4%
50–69 years	31.3%	30.2%
>69 years	7.2%	1.7%
Level of education		
Elementary and Medium school	18.7%	7.8%
High school	47.4%	50.2%
Degree	30.1%	35.2%
Post degree	3.9%	6.8%
Income		
High > 48,000 EUR/year	6.0%	2.3%
Medium High 30,001–48,000 EUR/year	38.3%	35.6%
Medium Low 15,001–30,000 EUR/year	43.3%	51.1%
Low < 15,000 EUR/year	12.4%	11.0%
Number of family members		
1 member	8.8%	3.8%
2 members	19.7%	13.7%
3 members	24.0%	26.2%
4 members	30.7%	37.6%
5 members	12.1%	15.6%
More than 5 members	4.5%	3.2%

Source: Authors elaboration.

**Table 3 foods-12-01661-t003:** Factorial analysis Rotated Component Matrix ^a^.

		Latent Factors Group			
		Sustainable Inclination (SI)	Consumption Experience (CE)	Stores (St)	Informed Consumers_4.0 (IC)
I believe that the shelf-life of bakery products can help reduce food waste and improve sustainability	Si1	**0.782**	0.002	−0.002	0.101
I believe that the label and green labels are important for understanding the bakery supply chains that support circular economy models and environmentally friendly production systems	Si2	**0.738**	0.299	−0.009	−0.129
For me, the reputation of companies and their social, ethical and sustainable responsibility are important	Si3	**0.679**	0.16	−0.031	0.238
The use of eco-sustainable and differentiable food packaging is one of the main ways to reduce pollution	Si4	**0.624**	0.174	−0.005	0.376
We need to protect biodiversity and safeguard local varieties	Si5	**0.590**	0.324	0.054	0.05
I use 0 km products for my experience and to support small local businesses	Ce1	0.195	**0.640**	0.05	0.027
It is important to support farmers with a fair price for quality and food safety guarantees	Ce2	0.171	**0.593**	−0.071	0.35
Clarity and transparency in the production technique and in the processing and marketing phase of food products are important for quality and safety	Ce3	0.286	**0.517**	−0.121	0.28
Eco-sustainable products with quality certification mark (PDO, PGI, Organic, etc.) affects on my food choices	Ce4	0.36	**0.482**	−0.047	0.043
When buying bakery products I try to pay attention to Food Safety and Quality (control of toxins, pathogens, pesticides, etc.)	Ce5	0.285	**0.632**	−0.04	−0.286
The price influences my choice and propensity to purchase bakery products	Ce6	−0.135	**0.622**	0.025	0.365
I choose to buy cookies from my favorite retailer	St1	−0.015	−0.031	**0.899**	0.05
I like the idea of buying salted snacks and other bakery products from different stores	St2	0.095	−0.083	**0.883**	−0.102
When I buy bread from my favorite retailer, if necessary, I can receive information and suggestions	St3	−0.08	0.038	**0.788**	0.125
The availability and possibility of buying online influence my food choices	Ic1	0.144	0.004	0.025	**0.767**
I follow social, advertising, food blogs and media that may influence me to buy sustainable food (including bakery products)	Ic2	0.186	0.338	0.098	**0.620**

^a^ Extraction Method: Principal Component Analysis. Rotation Method: Varimax with Kaiser Normalization. KMO Test: Kaiser–Meyer–Olkin Measure of Sampling Adequacy = 0.802. Bartlett’s Test of Sphericity: Approx. Chi-Square = 3391.38; df = 120; Sig. = 0. Source: Authors elaboration from data analysis in SPSS Ver. 20.

**Table 4 foods-12-01661-t004:** Standardized factor loading Composite Reliability (CR) Average Variance Extracted (AVE) and Cronbach’s Alpha.

	Factor and Item	Standardized Factor Loading	Composite Reliability (CR)	Average Variance Extracted (AVE)	Cronbach’s Alpha
	**Sustainable Inclination (SI)**		**0.848**	**0.528**	**0.779**
Si1	I believe that the shelf-life of baked goods can help reduce food waste and improve sustainability	0.702			
Si2	I believe that the label and green labels are important for understanding the bakery supply chains that support circular economy models and environmentally friendly production systems	0.729			
Si3	For me, the reputation of companies and their social, ethical and sustainable responsibility are important	0.756			
Si4	The use of eco-sustainable and differentiable food packaging is one of the main ways to reduce pollution.	0.742			
Si5	We need to protect biodiversity and safeguard local varieties	0.703			
	**Consumption Experience (CE)**		**0.792**	**0.491**	**0.650**
Ce1	I use 0 km products for my life and to support small local businesses	0.622			
Ce2	It is important to support farmers with a fair price for quality and food safety guarantees	0.763			
Ce3	Clarity and transparency in the production technique and in the processing and marketing phase of food products are important for quality and safety	0.759			
Ce4	Eco-sustainable products with quality certification mark (PDO, PGI, Organic, etc.) affects on my food choices	0.646			
	**Informed Consumers 4.0 (IC)**		**0.806**	**0.678**	**0.546**
Ic1	The availability and possibility of buying online influence my food choices	0.726			
Ic2	I follow social, advertising, food blogs and media that may influence me to buy sustainable food (including bakery products)	0.910			
	**Stores (St)**		**0.881**	**0.717**	**0.824**
St1	I choose to buy cookies from my favorite retailer	0.950			
St2	I like the idea of buying salted snacks and other bakery products from different stores	0.894			
St3	When I buy bread from my favorite retailer, if necessary, I can receive information and suggestions	0.669			

Source: Authors elaboration from data analysis in Smart-PLS3.

**Table 5 foods-12-01661-t005:** Discriminant validity—Fornell–Larcker criterion.

	(CE)	(IC)	(St)	(SI)
Consumption Experience (CE)	**0.700**			
Informed Consumers 4.0 (IC)	0.394	**0.823**		
Stores (St)	−0.093	0.081	**0.847**	
Sustainable_Inclination (SI)	0.576	0.369	0.004	**0.727**

Source: Authors elaboration from data analysis in Smart-PLS3.

**Table 6 foods-12-01661-t006:** Heterotrait–Monotrait results.

	(CE)	(IC)	(St)	(SI)
Consumption Experience (CE)	-			
Informed Consumers 4.0 (IC)	0.621	-		
Stores (St)	0.132	0.106	-	
Sustainable_Inclination (SI)	0.794	0.535	0.068	-

Source: Authors elaboration from data analysis in Smart-PLS3.

**Table 7 foods-12-01661-t007:** Summary convergent validity and internal consistency of constructs.

Hypothesis	Estimate (β)	*t.* Value	*p*-Value	Hypothesis	Conclusion
Consumption_Experience->Sustainable_Inclination	0.515	15.292	0.000 ***	H1	Supported
Informed Consumers->Consumption_Experience	0.404	12.796	0.000 ***	H5	Supported
Informed Consumers->Stores	0.081	1.620	0.105	H4	Rejected
Informed Consumers->Sustainable_Inclination	0.163	5.096	0.000 ***	H2	Supported
Stores->Consumption Experience	−0.126	2.540	0.011 **	H6	Supported
Stores->Sustainable_Inclination	0.039	0.849	0.396	H3	Rejected

** *p*-Value < 0.01; *** *p*-Value < 0.001 Source: Authors elaboration from data analysis in Smart-PLS3.

**Table 8 foods-12-01661-t008:** Structural model—Multicollinearity check (Variance Inflated factors—VIFs).

	(CE)	(St)	(SI)
Consumption Experience (CE)			1.206
Informed Consumers 4.0 (IC)	1.007	1.000	1.204
Stores (St)	1.007		1.026

Values below 3.3 indicate an acceptable level of correlation among constructs. Source: Authors elaboration from data analysis in Smart-PLS3.

**Table 9 foods-12-01661-t009:** Construct cross-validated Communality.

	Q2
Consumption Experience (CE)	0.167
Informed Consumers 4.0 (IC)	0.120
Stores (St)	0.454
Sustainable_Inclination (SI)	0.293

An acceptable model has Q2 values above 0, which can be small (0.02), medium (0.15) or large (0.35). Source: Authors elaboration from data analysis in Smart-PLS3.

**Table 10 foods-12-01661-t010:** Percentage variation of favorite points of sale by consumers and spending frequency during the week interviewed before and during the health emergency from COVID-19. Source: The authors.

Favorite Points of Sale by Consumers					% Change
	Before COVID-19		During COVID-19		
MMR	306	64.6	238	50.2	−22.2
Local market	28	5.9	0	0.0	−100.0
Discount	66	13.9	65	13.7	−1.5
Proximity retail	40	8.4	129	27.2	222.5
Market online	15	3.2	21	4.4	40.0
Direct sale	19	4.0	21	4.4	10.5
**Spending frequency during the week**					
Once a week	121	25.5	377	79.5	211.6
Two-three times a week	268	56.5	87	18.4	−67.5
Four or more times a week	85	17.9	10	2.1	−88.2

## Data Availability

Eurostat: https://ec.europa.eu/eurostat/web/products-eurostat-news/-/ddn202209191; accessed on 13 January 2023 ISMEA: (https://www.ismeamercati.it/flex/cm/pages/ServeBLOB.php/L/IT/IDPagina/10464) accessed on 13 January 2023; eAmbrosia: https://ec.europa.eu/info/food-farming-fisheries/food-safety-and-quality/certification/quality-labels/geographical-indications-register/ accessed on 13 January 2023.

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
