# Peer review of "Habits, Health and Environment in the Purchase of Bakery Products: Consumption Preferences and Sustainable Inclinations before and during COVID-19"

_foods, 2023, doi:10.3390/foods12081661_

Round 1

Reviewer 1 Report

This study investigates the effects of habits, health and environment on the purchase of bakery products before and during COVID-19. It is an interesting study to compare and study the change of consumers' attitude when COVID strikes. Below are some comments for the authors to consider in improving the readability of this article.

1) Abstract: Please include a sentence to summarize the findings at the end of the abstract.

2) Technical: line 30, can put the references as [4-10]. Please kindly check through the entire manuscript. Eg line 452 etc

3) Introduction & Bakery products: The background is slightly long but the problem statement that associate it to the sustainability may need more elaborations. Try to describe the pre-COVID consumers' behavior apart from their behaviors during COVID. 

4) Survey method: for part 1 was conducted during 2017 (physical interviews) and for part 2 was during 2020 (online interview). Is there any prescreening criteria for the respondents? Will the sample size (different for part 1 and part 2) and survey method (physical vs online) affect the reliability of the data? 

line 232, traceability and traceability?

5) Data analysis. line 307 different size (...)? Missing data?

6) For all figures and tables, they were stated as source: the authors. Please include the related references. 

7) line 338, 342, 344. informed 4.0 meaning? It is slightly confusing and would be good to explain to guide the readers.

8) Syntax errors in Table 2

Eg ; Max 89 years 

Average 39 years

high oltre??

low fino a ??

Please check through the entire manuscript for similar minor errors.

line 367 38.5% e Young??

line 376 are an age?

line 387 variables; In? Please put a full stop instead of ; Please check through the entire manuscript for this error.

line 442 PLS-SEM

line 451 0, 7?

line 622 green experience and experience and ?

8) Discussion and Conclusion: Try to relate back to present study's objective and hypothesis, especially about sustainability. 

Reviewer 2 Report

Dear Authors,

In the introduction, the author needs to write and edit to clarify that The author must draft and modify the introduction to make it clear that the key contributions are to the study of the compared role.

The author needs to restructure the research hypothesis to focus on the theoretical argument instead of listing the studies.

The methodology needs to be more clearly stated. The author needs to update some of the following studies for both the overview and the methodology.

The author needs to add more discussion that needs to be supplemented because it is too sketchy.

The author needs to check Q2, f2 and multicollinearity of the model.

Kind regards,

Reviewer 3 Report

The paper is potentially interesting however it almost reads like like an opportunity lost. There is an extensive introduction to bakery products (which could potentially be condensed) but given the COVID dimension there is insufficient discussion of the impact of COVID on tastes and purchasing both with respect to food and its wider implications for sustainability. This is an important element as not only would it better introduce the research but such literature is needed to improve the discussion and conclusions which is currently quite weak and needs to be better connected to the literature on sustainability and the effects of COVID

Round 2

Reviewer 3 Report

The revisions have sufficiently improved the paper

Author Response

The authors thanks to appreciate the improvement of the paper.